# Advances in Production of Hydroxycinnamoyl-Quinic Acids: From Natural Sources to Biotechnology

**DOI:** 10.3390/antiox11122427

**Published:** 2022-12-09

**Authors:** Egle Valanciene, Naglis Malys

**Affiliations:** 1Bioprocess Research Centre, Faculty of Chemical Technology, Kaunas University of Technology, Radvilėnų pl. 19, LT-50254 Kaunas, Lithuania; 2Department of Organic Chemistry, Faculty of Chemical Technology, Kaunas University of Technology, Radvilėnų pl. 19, LT-50254 Kaunas, Lithuania

**Keywords:** hydroxycinnamoyl-quinic acids, chlorogenic acid, antioxidants, extraction, synthesis, biosynthesis by engineered micro-organisms

## Abstract

Hydroxycinnamoyl-quinic acids (HCQAs) are polyphenol esters formed of hydroxycinnamic acids and (-)-quinic acid. They are naturally synthesized by plants and some micro-organisms. The ester of caffeic acid and quinic acid, the chlorogenic acid, is an intermediate of lignin biosynthesis. HCQAs are biologically active dietary compounds exhibiting several important therapeutic properties, including antioxidant, antimicrobial, anti-inflammatory, neuroprotective, and other activities. They can also be used in the synthesis of nanoparticles or drugs. However, extraction of these compounds from biomass is a complex process and their synthesis requires costly precursors, limiting the industrial production and availability of a wider variety of HCQAs. The recently emerged production through the bioconversion is still in an early stage of development. In this paper, we discuss existing and potential future strategies for production of HCQAs.

## 1. Introduction

HCQAs (Table 1) are mostly produced in plants by ester formation of a hydroxycinnamic acid (primarily *p*-coumaric, caffeic, ferulic and sinapic acids) with a (-)-quinic acid from the phenylpropanoid pathway, and they are linked with lignin synthesis [1]. They belong to a large and diverse group of phenolic compounds, often termed as chlorogenic acids [2]. The extended list of chlorogenic acids, which contains approximately 400 compounds [2], also encompass several derivatives and isomers of quinic acid, including shikimic acid, its epimers, 4-deoxy-, *muco*-, methyl- and butyl-quinic acids esterified with hydroxycinnamic and hydroxybenzoic acids or some of their derivatives. The nomenclature and trivial names of compounds of chlorogenic acids are rather complicated and they are explained in [3].

Here, we primarily focus on the HCQAs (Table 1). Amongst these, the most abundant and important compound is 5-caffeoylquinic acid (5-CQA), often referred to as chlorogenic acid. In 2018, the market size of 5-CQA was 130 million and it is expected to reach 150 million US$ by 2025 [4].

Over the last decade, a wide interest in HCQAs has been reflected by an exponentially growing number of scientific publications (Figure 1). However, the main focus has been on HCQAs extraction from plants and to some extent on their chemical synthesis, whereas studies on microbial production of these acids have been limited.

HCQAs are used in pharmaceuticals, cosmetics, foods due to their therapeutic properties, such as antioxidant [5], anticancer [6], antimicrobial [7,8], antiobesity [9], hepatoprotective [10], antiviral and anti-inflammatory [11], antihypertensive [12] and neuroprotective [13]. The hydroxycinnamoyl moiety in these compounds determines the antimicrobial activity [14,15]. The strength of this property increases with the number of these moieties in the molecule. HCQAs exhibit antimicrobial activity against many bacteria, including *Enterococcus faecium*, *Escherichia coli* and others [16]. However, some cider yeasts (*Lactobacillus collinoides*, *Lactobacillus paracollinoides*) [17,18], fungi (*Aspergillus niger* C23308, *Fusarium graminearum* [19,20], *Fusarium culmorum* and *Fusarium graminearum sensu stricto* [21]) and lactic acid producing bacteria (*Lactobacillus johnsonii* NCC 533) [17,22] are resistant to 5-CQA due to their ability to catabolize this compound.

5-CQA has been found to be useful for diverse applications. Recently, it has been shown to have potential to counteract SARS-CoV-2 by reducing viral attachment to the host cell-surface heat shock protein A5 (HSPA5) [23]. 5-CQA stimulates short-chain fatty acid production in bacteria. The fermentation products of this acid stimulate the proliferation of *Bifidobacterium* spp. causing the decreased ratio between *Firmicutes* and *Bacteroidetes* [24]. The final compounds formed from 5-CQA and other HCQAs are hippuric acid and 3-hydroxyhippuric acid, which are used as non-specific biomarkers for polyphenol uptake or metabolism [25,26,27].

Moreover, 5-CQA has found applications in the synthesis of metals’ nanoparticles as a reducing and stabilizing agent [28,29]; in the production modification of carbon ceramic electrode for NADH detection [30]; in red food dye preparation from coupling of tryptophan and 5-CQA [31]; and the preservation of food by preparing the edible coating with chitosan [32]. This compound is also referred to as the major compound from instant coffee extract responsible for graphene green production from graphite and functionalization [33]. 5-CQA is considered as the precursor of caffeic and quinic acids, which may be obtained via hydrolysis reactions during extraction from plant material [34].

Importantly, the derivatives of HCQAs are also widely researched for their potential use in the production of drugs or the use in biopharmaceuticals [35,36]. Moreover, the extract of green coffee beans containing a large amount of 5-CQA has been certified by COSMOS to obtain a label of a natural raw material for cosmetics [37]. It is also produced by Naturex under the trade name Svetol^®^ as a supplement for body weight loss [38].

This paper aims to discuss and assess the recent advances in chemical synthesis of HCQAs, their extraction from plant material and agricultural waste as well as emerging bioproduction of these compounds using natural or modified micro-organisms. The HCQAs production strategies and relevant research developments are summarized.

## 2. HCQAs Extraction from Plants and Agricultural Waste

### 2.1. HCQAs Biosynthesis in Plants

The phenylpropanoid pathway forms a platform for HCQAs biosynthesis in plants with the main biochemical reactions presented in Figure 2. HCQAs can be synthesized from *p*-coumaric acid or *p*-coumaroyl-CoA by four different routes. The first route (blue arrows in Figure 2) requires the direct conversion of *p*-coumaric acid into other hydroxycinnamic acids (caffeic, ferulic, sinapic acids) via hydroxylation or methylation reactions mediated by *p*-coumarate 3-hydroxylase (C3H), caffeic/5-hydroxyferulic acid O-methyltransferase (COMT) or ferulic acid 5-hydroxylase (F5H) [39]. Then, hydroxycinnamic acids CoA esters are formed with mediation of 4-coumaroyl-CoA ligase (4CL) [40] and HCQAs are produced by transesterification reaction with quinic acid catalyzed by HCT/HQT [41]. The second route (pink arrows in Figure 3) is the conversion of hydroxycinnamates into glucosides and transesterification with quinic acid [41]. The third and fourth routes starts from *p*-coumaroyl-CoA, which is converted into *p*-coumaroyl-quinic (quinate shunt, red arrows in Figure 2), or *p*-coumaroyl-shikimic (shikimate shunt, green arrows in Figure 2) acids with additional conversion of obtained shikimate into ester of CoA and following esterification with quinic acid, respectively [39,41,42]. 3-CQA, 4-CQA are produced from 5-CQA, and 3,4-diCQA or 4,5-diCQAs are synthesized from 1,5-diCQA with mediation of isomerase or hydroxycinnamoyltransferase, respectively [43,44]. The dual activity of HQT enzyme results in 5-HCQAs production in cytosol and in the production of 3,4-diCQA, 3,5-diCQA or 4,5-diCQAs from two molecules of 5-CQA (acting as acyl-donor and acyl-acceptor) in the vacuoles [45]. In addition, 3,5-diCQA can be obtained from 5-CQA and caffeoyl-CoA with mediation of HCT and ICS enzymes [46]. *Cis* isomerization may occur with mediation of the non-specified/non-identified enzyme [47] or due to the exposure to ultraviolet irradiation [48]. To date, the major synthesis routes in plants are shikimic or quinic acids shunts, which results in the production of *p*-CoQAs, CQAs or FQAs [39,44,49,50].

#### 2.1.1. Plant Sources of HCQAs 

All types of HCQAs (from Table 2) may be found in plants (Figure 3) and they are mainly localized in fruit skin, seeds, kernels, leaves, or husks. The richest sources of HCQAs (mainly mono- and dicaffeoylquinic acids) are Yerba mate, white, green teas and coffee (Table 2). Although coffee is considered a major 5-CQA source, depending on the coffee type it may contain a lower total amount of 5-CQA but a larger variety of other HCQAs (including sinapoyl- derivatives such as 3-SQA, 4-SQA, 5-SQA, 3-S-5-CQA, 3-S-4-CQA, 4-S-3-CQA, 3-S-5-FQA, 3-F-4-SQA, 4-S-5-FQA) than teas [51,52,53]. The plants from *Asteraceae, Cichorium, Phaseoullus, Brassica, Solanaceae* and *Lamiaceae* families possess high amounts of different HCQAs [54,55,56]. Many fruits and berries are rich in CQAs with the highest total concentrations of 200–570 mg/kg wet biomass determined in cherry, quince, mulberry, bilberry, and sweet granadilla [57]. Food, crop and agro-industrial waste can be used as an alternative source for the bio-refinery of HCQAs [58,59]. Coffee by-products are some of the richest waste sources containing up to 10–23% of HCQAs [51,60]. Principally, all plants containing high levels of alkaloids possess significant amounts of HCQAs [61,62].

#### 2.1.2. Marine Sources of HCQAs

HCQAs are present in marine sources, mainly microalgae (*Spongiochlori* sp., *Euglena cantabrica*, *Anabaena doliolum*, *Porphyra tenera*, *Undaria pinnatifia*) and cyanobacteria (*Nostoc* sp.) [64,65,66,67]. The HCQAs synthesis pathway in these organisms is similar to higher plants [68]. Although microalgae and cyanobacteria contain high levels of extractable phenolic compounds (phenolic acids) and quinic acid, the abundance of various HCQAs is limited. To date, the data on abundance of 5-CQA is only available for some species of marine plants and cyanobacteria. Its concentration reaches up to 78 µg/g DW for microalgae and 9.55 µg/g DW for cyanobacteria (Table 3). The higher amount of 5-CQA is determined in algae due to the adaptation to abiotic and biotic stress occurring in the evolutionary advanced micro-organisms [65].

### 2.2. HCQAs Extraction

HCQAs are usually extracted by conventional, solid–liquid extraction (SLE) and nonconventional or intensified techniques, such as ultrasound assisted (UAE), microwave assisted (MAE), pressurized liquid (PLE), supercritical fluid (SFE), enzymatic extraction (EAE) [69,70] or infrared assisted extraction (IAE) [71] (Table 4). Only in EAE, the release and extraction is performed for HCQAs trapped within the plant cell walls using enzymes (cellulases, glucosidases, proteases, dextranases, xylanases and ligninolytic enzymes that do not hydrolase the HCQAs into their constituents) in aqueous buffers or ionic liquids [72,73,74,75]. The non-conventional or intensified extraction methods are considered to be more efficient. In extraction process, parameters (such as temperature, time, pH of solvents, particle size, solvent type and concentration, its volume and other specific parameters of the process (e.g., microwave power, ultrasonic frequency, enzyme concentration)) can be optimized [69,76]. The final yield of HCQAs depends on the raw material as well as the extraction method (Table 4). The highest yields (up to 4% of raw material) of HCQAs are determined for the coffee by-products, honeysuckle (*Lonicera japonicae)* and its flowers or by-products, sunflower seed kernels [60,77,78,79,80,81,82]. All the extraction methods are potentially suitable for the HCQAs except for SFE, which is considered as the green extraction technique, but the recovery yields do not exceed~52%, due to reduced solubility in the nonpolar supercritical fluid [80,83]. The MAE and IRE are usually very efficient and require a short time for the extraction process, but the large size microwave extractors in industry remain too expensive. Therefore, SLE is considered the most relevant method for industrial application due to its simplicity, reproducibility, low cost and possibility to use environment friendly solvents [84]. Intensified methods are mostly applied for the recovery of HCQAs from waste [71,85,86,87,88,89,90,91,92,93] (Table 4).

Many intensified extraction methods have been adopted for waste valorization. Furthermore, combined processes, such as microwave-assisted simultaneous distillation and dual extraction (MSDDE) [90], multi-frequency multimode modulated vibration (acoustic probe) technique (MMM) [91] or simultaneous SLE extraction and solid-state fermentation (SLE-SSF) [92] have been developed for 5-CQA extraction (Table 4).These three methods are more effective than SLE or other non-conventional extraction methods, however, MSDDE and MMM are too expensive for industrial application, and only SLE-SSF can be applied for large scale extraction and production of 5-CQA with high enrichment of raw material (for example, up to 400% for coffee pulp) [92].

For the separation, concentration and purification of HCQAs, mesoporous resins (recovery 65.03%, purity of CQAs 89.27%) [94], a high-speed countercurrent or high-performance liquid chromatography (recovery up to 99.56%) [82,89,95], fractional extraction in centrifuges (yield 35%, purity 99.5%) [96], isocratic system with three zones of simulating moving beds (0.04–0.2 g/L extract, purity 99.27%) [77], ultrafiltration membranes (rejection 92%) [97], molecularly imprinted polymer (recovery 60.08–72.59%, extraction yield 12.57 mg/g raw material) [79], imprinted magnetic nanomaterials or membranes (recovery 86–102%) [98,99], ethanol/salt aqueous two-phase system (95.76% maximum extraction) [100] are used. The chromatography-based purification is the most popular method, which is often used in industrial applications, due to its simplicity, effectiveness and reasonable cost. The fractional extraction in centrifuge or three-zone simulated moving bed method are based on continuous and selective processes, which allows for separating and concentrating 5-CQA with high purity (97.25–99.27%) [77,96]. However, only the fractional extraction in centrifuge is a cost-effective method and can be applied in industrial scale production [96]. The other separation and purification methods have been tested only in lab-scale production and may be limited due to some disadvantages, such as high cost, low effectiveness, low purity or selectivity, long analysis time, usage of toxic (organic) solvents, or membrane fouling [79,94,98,100,101,102,103].

The extraction and identification of HCQAs suffer from some limitations. Firstly, the stability of HCQAs is reduced by the isomerization or destruction reactions under extraction conditions, especially if these conditions are harsh [2,93]. Secondly, transesterification reactions can occur due to the activity of chlorogenate-dependent caffeoyltransferase, when the extraction of fresh plant material is performed in alcohol or alcohol-water mixtures [104]. The transformed products may be mistakenly identified as the new bioactive ingredients of the plant sources [105]. Another problem is the identification of the compounds in the extracts due to the limited number of commercially available standards and similar properties or spectra of HCQAs as summarized in [2,106].

Extraction and purification processes can be more environment-friendly if ionic liquids or eco-friendly natural deep eutectic solvents are used instead of organic solvents. They have a higher affinity to HCQAs than water [107], and it is easy to purify these solvents by distillation and to re-use them for extraction processes [108,109]. Furthermore, the use of ionic liquids can result in a shorter extraction time [75,110]. Generally, the recycling and re-use of the extraction solvents, application of high solid–to–liquid ratio, and reduced processing time helps to lower the total cost of the extraction [100,111].

The extraction of HCQAs results in large amounts of the residual plant material. Aiming for the sustainable circular economy and resource conservation, these residues can be utilized for combustion [112], as a biofuel (e.g., biodiesel, bioethanol, biochar and liquid pyrolysis product production) [113,114,115] or other applications [60,113,116,117,118,119,120,121,122,123] (Figure 4), which allows for reducing the solid waste amount and/or covering up to 100% of phytoremediation cost [124,125].

## 3. Chemical Synthesis of HCQAs

The major HCQAs containing hydroxycinnamoyl moieties can be chemically synthesized performing either esterification or condensation reaction between quinic and hydroxycinnamic acids using pyridine or DMAP as homogenous catalyst in organic solvents (dichlormethane or DMF). The reactions with the best total yields are presented in Figure 5.

The highest total yields have been obtained with Sefkow synthesis method for 1-CQA, 3-CQA, 4-CQA or 5-CQA isomers (Table 5) using esterification reaction between caffeoylchloride derivative and quinic acid derivatives with unprotected OH groups at 1, 3, 4 and/or 5 positions followed by 1–2 steps of hydrolysis reaction of protecting groups [126,127]. The non-protected quinic acid can also be used in the esterification with suitably protected hydroxycinnamoyl derivative, which results in the formation of all mono- substituted HCQAs with moieties at 1-, 3-, 4- and 5- positions [128]. The synthesis of 5-FQA can be obtained from quinic acid ester with malonate and vanillin without any protected hydroxyl group via Knoevenagel condensation reaction when the (*E*)-double bond is formed (the total yield of 19%) [129]. Independently of the synthesis method, the final crystalline HCQAs can be purified by recrystallization or by a complex procedure (extraction, concentration and chromatographic purification followed by recrystallization) for the improved purity of the final compound [126,127,129]. All the above methods are commonly used in manufacturing, and they are considered cost-effective as the solvents can be recovered by distillation and reused.

There are positive and negative aspects of condensation and esterification methods. Higher total yields can be achieved with the esterification method compared to the condensation method. Irrespective of method, the synthesis of di- or tri-HCQAs is more complex than that of mono-HCQAs. Esterification reaction requires the protection/deprotection of active hydroxyl groups because the HCQAs molecules are sensitive to basic and strong acidic or hydrogenative conditions due to the double bond reduction, transesterification or isomerization reactions as well as possible cleavage reactions [129,131,132]. Additionally, esterification may require low or ultralow temperature, solvent system CH_2_Cl_2_/pyridine/DMAP ratio optimization, and long acidic hydrolysis time for the deprotection reaction [126,127,131,132,136], which could cause the decreased yields of HCQAs due to the side reactions. In contrast, the condensation reaction by Knoevenagel involves benzyl aldehydes use of which does not require any protection of reactive hydroxyl groups [137]. 1-HCQAs and di- or tri- substituted HCQAs are more difficult to obtain by both condensation and esterification methods due to increased steric hindrance.

The prices of the major chemical substances for both synthesis methods are presented in Table 6. The Knoevenagel condensation reaction requires benzyl aldehydes, which are currently commercially produced from naphtha. Hydroxycinnamic acids for the esterification reaction are expensive as they are produced mainly by chemical synthesis. Even biotechnological production of vanillin, syringaldehyde or extraction of hydroxycinnamic acids from plant biomass could not reduce the high synthesis costs of HCQAs.

## 4. Biosynthesis of HCQAs in Non-Modified and Modified Micro-Organisms

### 4.1. Non-Modified Micro-Organisms

Non-modified micro-organisms, such as bacteria and fungi are able to produce 5-CQA from organic carbon (Table 7). The detectable amounts of 5-CQA have been observed with bacteria such as *Brevibacillus borstelensis*, *Bacillus amyloliquefaciens*, *Bacillus badius*, *Sphingomonas yabuuchiae*, *Enterobacter tabaci*, *Paenibacillus phoenicis* and fungi including *Colletotrichum acutatum*, *Lodderomyces elongisporus*, *Sphingomonas yabuuchiae*, *Enterobacter tabaci*, *Paenibacillus phoenicis* [139], *Sordariomycetes* sp. [140], *Penicillium flavigenum*, *Screlotium rolfsii* [141,142]. *L. elongisporus* sp. S216 and *P. flavigenum* (CML2965) exhibit the 5-CQA titers that could be of interest for industrial scale production [139,141] (Table 7). The pathway of 5-CQA biosynthesis in micro-organisms is considered to be similar to plants [139]. Interestingly, an ortho-adipate pathway enabling production of 1-CQA and 3,4,5-triCQA may be functional in *Streptomyces albogriseolus* KF977548, which has been isolated from decaying wood [143].

Micro-organisms producing 5-CQA can be used for the valorization of agricultural waste. For example, *B. amyloliquefaciens* B17 has been applied for the successful fermentation of mango peels in liquid state fermentation at 37 °C [145]. Further research is required to screen and develop micro-organisms for the valorization of different types of waste.

### 4.2. Modified Micro-Organisms

The HCQAs biosynthesis pathways have been developed in *E. coli*, *Saccharomyces cerevisiae*, and *Pichia pastoris*. For the production of 5-CQA from caffeic acid, the *E. coli* has been engineered by introducing hydroxycinnamoyl-CoA quinate transferase gene *HQT* from *Nicotiana tabacum* and 4-coumarate CoA:ligase gene 4CL, which mediates the formation of coenzyme A thioester with hydroxycinnamic acids and deletion of *aroD* gene, which is required for the conversion of 3-dehydroquinate into 3-dehydroshikimate [145]. This recombinant strain B-101 was able to accumulate 3-dehydroquinate and caffeoyl-3-dehydroquinate, but it was not able to produce 5-CQA in higher concentrations than 16 mg/L. When gene *ydiB* encoding shikimate/quinate dehydrogenase was overexpressed followed by the vector and cell concentration optimization, then the production up to 450 mg/L in 24 h was achieved from quinate and additionally supplied caffeic acid [145].

Recently, the synthesis of 5-CQA and p-coumaroyl shikimate by expressing shikimate gene modules in *E. coli* has been demonstrated. The overexpression of five genes of the shikimate pathway (*ppsA*, *tktA, aroGf, aroB, ydiB*) and heterologous genes *TAL*, *HpaBC*, *4CL*, *HST*, and *HQT* resulted in the biosynthesis of 109.7 mg/L 5-CQA from glucose [146] (Figure 6).

The polyculture technique has been applied for 5-CQA production using engineered *E. coli*. In the two culture technique, *E. coli* B-102 was inoculated in a filtrated culture mixture of modified *E. coli* strain B-TP-CA2, which produced caffeic acid from glucose [147]. After 45 h the highest concentration (78 mg/L) of 5-CQA was obtained. Similarly, de novo 5-CQA production using the polyculture technique with three recombinant *E. coli* strains containing biosynthetic modules of caffeic acid, quinic acid and 5-CQA, reached 250 µM (or ~88 mg/L) concentration after 18 h of incubation [148]. In both cases, a rather low yield of targeted compound could be acceptable due to the usage of a low-cost primary compound (glucose).

Yeast has also been engineered for the synthesis of HCQAs. The expression of tobacco *4CL* and globe artichoke *HCT* genes in yeast *Saccharomyces cerevisiae* resulted in the formation of N-(*E*)-*p*-coumaroyl-3-hydroxyanthranilic acid as a primary product, which is similar to avenanthramides [149]. Subsequently, a successful biosynthesis of 5-CQA and 5-pCoQA in yeast was achieved by expressing a BAHD enzyme *Nt*HQT from tobacco *Nicotiana tabacum* and 4CL5 [150]. Recently, it has been shown that GDSL lipase-like ICS enzyme from *Ipomoea batatas* can be used for the efficient conversion of 5-CQA into 3,5-diCQA in *P. pastoris* [46].

Further improvements have been achieved by introducing into *S. cerevisiae* the de novo 5-CQA synthesis pathway, including PAL2, C4H, 4CL1, C3′H, CPR1, CPR2, HQT2, YdiB, CYB5 and implementing the following modifications (Figure 6): (1) unlocking the shikimate pathway and optimizing carbon distribution by overexpressing the L-phenylalanine feedback-insensitive DAHP synthase (ARO3^K222L^), L-tyrosine feedback-insensitive DAHP synthase (ARO4^K229L^), pyruvate kinase 1 mutant with reduced catalytic activity (PYK1^D146N^) and transketolase (TKL1); (2) optimizing the L-phenylalanine branch and pathway balancing by overexpressing the l-tyrosine feedback-insensitive chorismate mutase (ARO7^G141S^), endogenous prephenate dehydratase (PHA2), NADH kinase (POS5), and cytochrome b5 (Cyb5); (3) increasing the copy number of 5-CQA pathway genes encoding hydroxycinnamoyl-CoA quinate transferase 2 (HQT2) and cytochrome P450 98A3 (C3′H) [151]. The engineered *S. cerevisiae* strain produced 5-CQA to 234.8 ± 11.1 mg/L in shake flask culture and 806.8 ± 1.7 mg/L in fed-batch fermentation [151].

## 5. Conclusions

HCQAs are becoming very important compounds in cosmetics, medicine and food supplements production due to their outstanding properties. Therefore, the need of HCQAs, especially 5-CQA, is constantly increasing. The main strategies to obtain these compounds are based on their extraction from plant biomass and chemical synthesis. Recently, biosynthesis using modified or non-modified micro-organisms has attracted significant research efforts.

Several conventional and intensified methods have been developed for extraction of HCQAs from plants or marine biomass. Amongst the most promising are solid–liquid extraction techniques (SLE), pressurized liquid (PLE), supercritical fluid (SFE), enzymatic extraction (EAE), even if SLE remains dominant. The successful application of other intensified methods, such as microwave-assisted simultaneous distillation and dual extraction (MSDDE), multi-frequency multimode modulated vibration (acoustic probe) technique (MMM) or simultaneous SLE extraction and solid-state fermentation (SLE-SSF), was demonstrated with some evident success for the 5-CQA. Their suitability for other HCQAs remains to be explored. Despite that, intensified methods can result in better yields and can require less time; however, they are expensive, which limits their industrial application. 5-CQA extraction has already reached the stage of industrialization. 5-CQA is mainly produced (>75%) from honeysuckle, eucommia and green coffee bean. For the less abundant HCQAs, such as SQAs (which is present at extremely low yields in plants), further improvements of the extraction methodology are required. 

Chemical synthesis enables achievement of moderate yields of mono-, di- or tri- HCQAs. It requires large quantities of hydroxycinnamic acid and is based on use of condensation or esterification methods. Esterification enables achievement of a better yield, but the high costs of reagents limits use of this method. Overall, the chemical synthesis of HCQAs is expensive and environmentally unfriendly because the primary compounds of the synthesis are obtained mainly from naphtha and its refinery products. Moreover, the use of halogenized organic compounds generates toxic waste. Despite this, this method could be preferable for the production of naturally scarce SQAs or to synthesize HCQAs that are not obtainable by other methods.

Biotechnological production of HCQAs is considered a most promising approach that enables consolidation of green chemistry and circular economy objectives. Despite its early stage, several non-modified micro-organisms have already been shown to produce the CQAs to yields that are comparable to those obtained by extraction from plant biomass. Significantly, the engineered bacteria *Escherichia coli* or yeast *Saccharomyces cerevisiae* have been developed to synthesize a greater variety of HCQAs, such as 5-CQA, 5-FQA, 5-*p*-CoQA from simple carbon sources. The 5-CQA titer of approximately 0.8 g/l has been achieved in fed-batch fermentation using *Saccharomyces cerevisiae*. Although the biotechnological production of HCQAs requires a significant advancement to make it suitable for industrial application, the utilization of micro-organisms shows great promise in the recycling and recovery of HCQAs from organic waste.

## Figures and Tables

**Figure 1 antioxidants-11-02427-f001:**
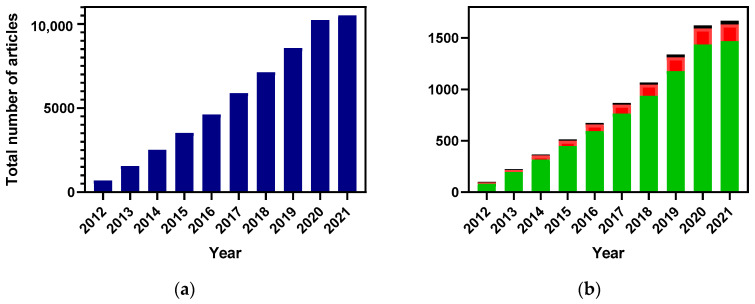
The number of publications dedicated to HCQAs research over the last 10 years (data based on information retrieved from scopus.com on 22 February 2022). Number of publications is represented as following: total ((**a**) blue bar), HCQAs extraction from plants ((**b**) green bar), HCQAs chemical synthesis ((**b**) red bar), and HCQAs production in micro-organisms ((**b**) black bar).

**Figure 2 antioxidants-11-02427-f002:**
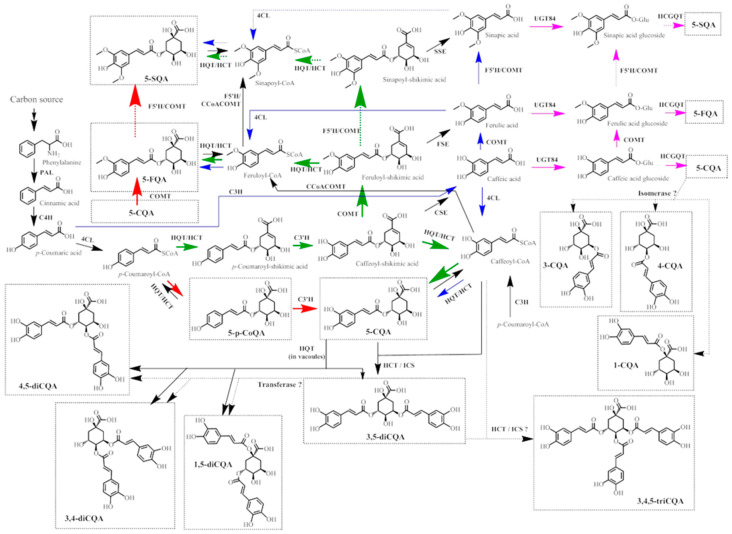
Pathways for the biosynthesis of HCQAs in plants. The four main routes of phenylpropanoid metabolism are highlighted in different colors: green, shikimate shunt; red, quinate shunt; blue and pink, direct conversion and cinnamoyl glucosides pathway. Dashed arrows show the suggested enzymatic reactions. Abbreviations: PAL, L-phenylalanine ammonia-lyase; C4H, cinnamate 4-hydroxylase; 4CL, *p*-coumaroyl-CoA ligase; HCT/HQT, 4-hydroxycinnamoyl CoA - shikimate/quinate hydroxycinnamoyl transferase; C3’H, *p*-coumaroyl shikimate/quinate 3’-hydroxylase, CSE - caffeoyl shikimate esterase, ICS isochlorogenate synthase, HCT - hydroxycinnamoyl-CoA shikimate/quinate hydroxycinnamoyltransferase; HQT, hydroxycinnamoyl-CoA quinate hydroxycinnamoyl transferase; CSE, caffeoyl shikimate esterase; FSE, feruoyl shikimate esterase; SSE, sinapoyl shikimate esterase; COMT, caffeic/5-hydroxyferulic acid O-methyltransferase; C3H, *p*-coumarate 3-hydroxylase (ascorbate peroxidase); CCoAOMT, caffeoyl-CoA 3-O-methyltransferase; UGT84, UDP-glucoside transferase; HCGQT, hydroxycinnamoyl D-glucose:quinate hydroxycinnamoyl transferase; F5’H—ferulic acid 5-hydroxylase.

**Figure 3 antioxidants-11-02427-f003:**
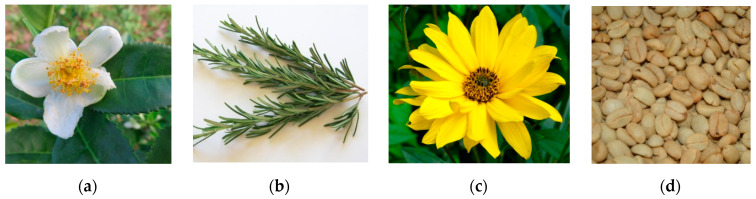
Some natural sources of HCQAs: tea tree (*Camellia sinensis)* (**a**), rosemary (*Rosmarinus officinalis*) (**b**), mountain arnica (*Arnica montana*) (**c**), coffee beans (*Coffea* sp.) (**d**).

**Figure 4 antioxidants-11-02427-f004:**
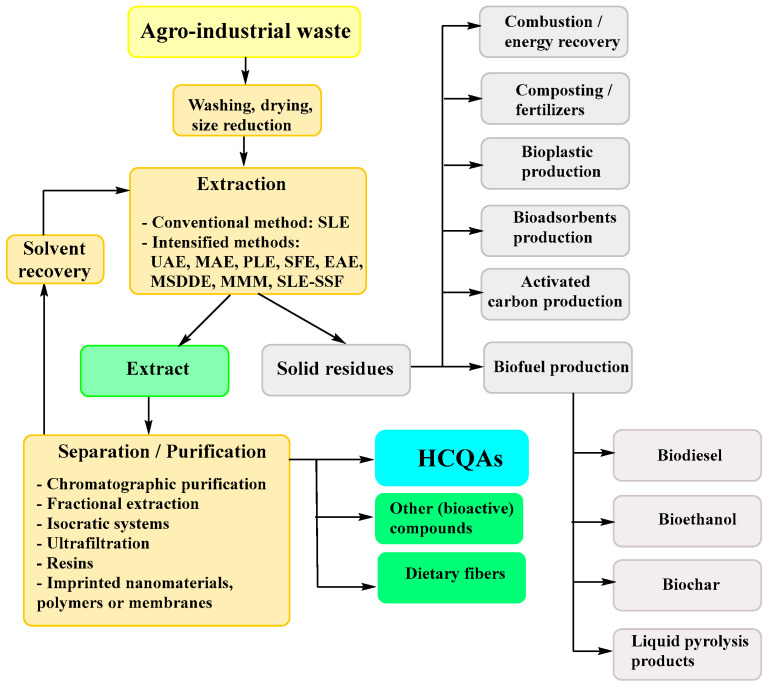
General agro-industrial waste treatment possibilities. Abbreviations: EAE—enzyme assisted extraction; MAE—microwave assisted extraction; MMM—multi-frequency multimode modulated vibration (acoustic probe) technique; MSDDE—microwave-assisted simultaneous distillation and dual extraction; PLE—pressurized liquid extraction; SFE—supercritical fluid extraction; SLE—solid-liquid extraction techniques; SLE-SSF—simultaneous SLE extraction and solid-state fermentation; UAE—ultrasound assisted extraction.

**Figure 5 antioxidants-11-02427-f005:**
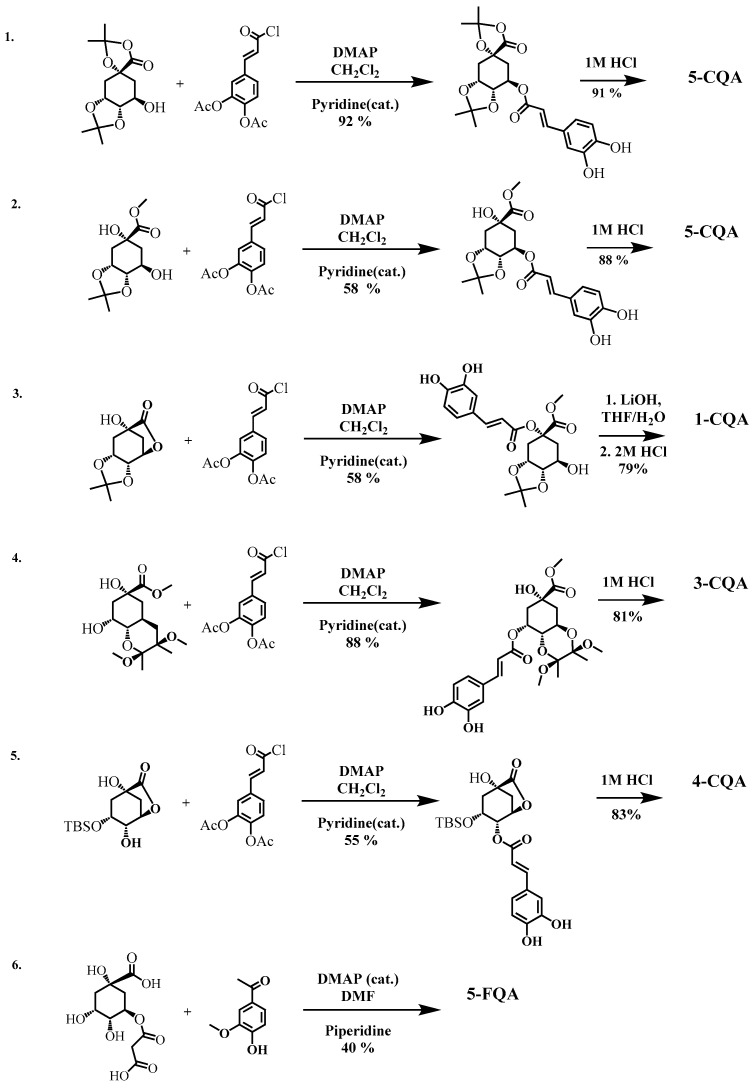
Chemical synthesis of 1-CQA, 3-CQA, 3-CQA and 5-CQA. Selected reactions with highest yields reported by [127] (1); [130] (2); [126] (3–5); and [129] (6). All reactions performed at room temperature.

**Figure 6 antioxidants-11-02427-f006:**
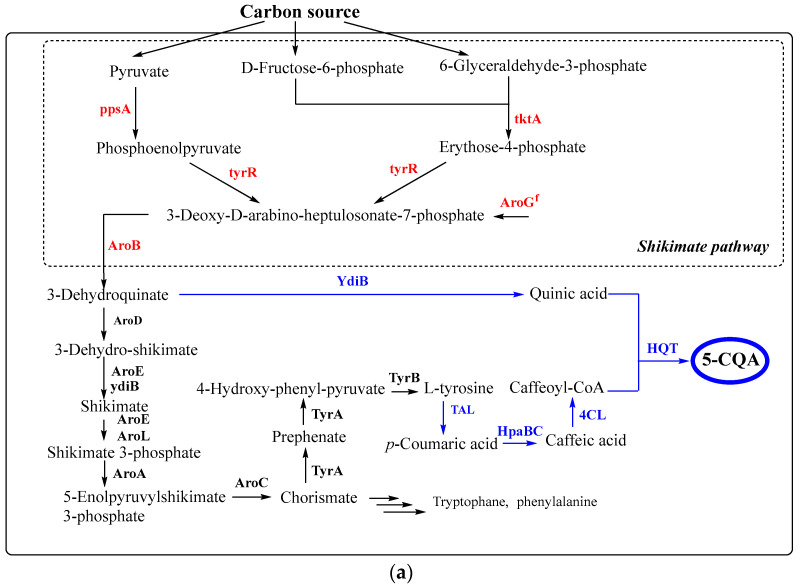
The synthesis of chlorogenic acid in *E. coli* (**a**) and *S. cerevisiae* (**b**) from carbon source according to [146; 151]. Black and blue arrows correspond to native and non-native pathways, respectively. Dashed arrows represent the complex processes. The blue and red names of genes correspond to inserted and overexpressed genes, respectively. Abbreviations: AroH - phospho-2-dehydro-3-deoxyheptonate aldolase; TyrR—transcriptional regulatory protein; AroF - phospho-2-dehydro-3-deoxyheptonate aldolase; AroG—phospho-2-dehydro-3-deoxyheptonate aldolase; AroD—5-dehydroquinate dehydratase; AroB—dehydroquinate synthase; *PAL2*—phenylalanine ammonia lyase from *Arabidopsis thaliana; C3*′*H*—cytochrome P450 98A3 from *A. thaliana; CPR1* and *AtCPR2*—P450 reductases from *A. thaliana;* YdiB—quinate/shikimate dehydrogenase from *E. coli*; *AtC4H*, cinnamate-4-hydroxylase from *A. thaliana;* 4CL—4-coumarateCoA:ligase from *Oryza sativa; 4CL1*, 4-coumarate:CoA ligase 1 from *A. thaliana;* HQT—hydroxycinnamoyl-CoA quinate transferase from *Nicotiana tabacum;* HQT2—hydroxycinnamoyl-CoA quinate transferase 2 from *Cynara scolymus*; *ARO3^K222L^*—l-phenylalanine feedback-insensitive DAHP synthase; *ARO4^K229L^*—l-tyrosine feedback-insensitive DAHP synthase; *ARO7^G141S^*—l-tyrosine feedback-insensitive chorismate mutase; *PYK1^D146N^*—pyruvate kinase 1 mutant with reduced catalytic activity.

**Table 1 antioxidants-11-02427-t001:**
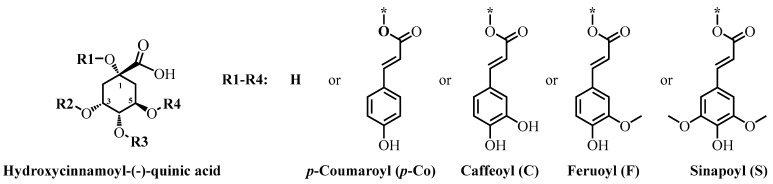
The principal chemical structure of HCQAs.

Abbreviation	Chemical Name of Compound (*Trivial Name*)	R1	R2	R3	R4
5-CQA	5-caffeoylquinic acid (*chlorogenic acid*)	H	H	H	C
5-FQA	5-feruloylquinic acid	H	H	H	F
5-*p*-CoQA	5-*p*-coumaroylquinic acid	H	H	H	*p-*Co
5-SQA	5-sinapoylquinic acid	H	H	H	S
4-CQA	4-caffeoylquinic acid (*cryptochlorogenic acid*)	H	H	C	H
4-FQA	4-feruloylquinic acid	H	H	F	H
4-*p*-CoQA	4-*p*-coumaroylquinic acid	H	H	*p-*Co	H
4-SQA	4-sinapoylquinic acid	H	H	S	H
3-CQA	3-caffeoylquinic acid (*neochlorogenic acid*)	H	C	H	H
3-FQA	3-feruloylquinic acid	H	F	H	H
3-*p*-CoQA	3-*p*-coumaroylquinic acid	H	*p-*Co	H	H
3-SQA	3-sinapoylquinic acid	H	S	H	H
1-CQA	1-caffeoylquinic acid (*pseudochlorogenic acid*)	C	H	H	H
1-FQA	1-feruoylquinic acid	F	H	H	H
1-*p*-CoQA	1-*p*-coumaroylquinic acid	*p*-Co	H	H	H
1-SQA	1-sinapoylquinic acid	S	H	H	H
3,4-diCQA	3,4-dicaffeoylquinic acid	H	C	C	H
3,4-diFQA	3,4-diferuloylquinic acid	H	F	F	H
3,4-di-*p*-CoQA	3,4-di-*p*-coumaroylquinic acid	H	*p*-Co	*p*-Co	H
4-SQA	4-sinapoylquinic acid	H	H	S	H
5-SQA	5-sinapoylquinic acid	H	H	H	S
3-S-5-CQA	3-sinapoyl-5-caffeoylquinic acid	H	S	H	C
3-S-4-CQA	3-sinapoyl-4-caffeoylquinic acid	H	S	C	H
4-S-3-CQA	4-sinapoyl-3-caffeoylquinic acid	H	C	S	H
3-S-5-FQA	3-sinapoyl-5-feruloylquinic acid	H	S	H	F
3-F-4-SQA	3-feruloyl-4-sinapoylquinic acid	H	F	S	H
4-S-5-FQA	4-sinapoyl-5-feruloylquinic acid	H	H	S	F
4,5-diCQA	4,5-dicaffeoylquinic acid	H	H	C	C
3,5-diCQA	3,5-dicaffeoylquinic acid (*isochlorogenic acid A*)	H	C	H	C
3,5-diFQA	3,5-diferuloylquinic acid	H	F	H	F
3,5-di-*p*-CoQA	3,5-di-*p*-coumaroylquinic acid	H	*p*-C	H	*p*-Co
3-C-5-*p*-CoQA	3-caffeoyl-5-*p*-coumaroylquinic acid	H	C	H	*p*-Co
3-C-5-FQA	3-caffeoyl-5-feruloylquinic acid	H	C	H	F
3-C-5-SQA	3-caffeoyl-5-sinapoylquinic acid	H	C	H	S
1,5-diCQA	1,5-dicaffeoylquinic acid (*cynarin)*	C	H	H	C
1,4-diCQA	1,4-dicaffeoylquinic acid	C	H	C	H
3,4,5-triCQA	3,4,5-tricaffeoylquinic acid	H	C	C	C
1,3,5-triCQA	1,3,5-tricaffeoylquinic acid	C	C	H	C
4-F-5-CQA	4-feruloyl-5-caffeoylquinic acid	H	H	F	C

**Table 2 antioxidants-11-02427-t002:** Plant sources with the most abundant HCQAs available.

Source	HCQAs, g/kg Dry Weight											Ref.
	3-CQA	4-CQA	5-CQA	3,4-diCQA	3,5-diCQA	4,5-diCQA	5-FQA	3-*p*-CoQA	4-*p*-CoQA	5-*p*-CoQA	Total	
Leaves and Thalli of Yerba mate *(Ilex paraguariensis*)	27.6	6.80	12.07	5.82	29.59	9.98	n.d.	n.d.	n.d.	n.d.	91.89	[54]
Leaves of White tea (*Camellia sinensis*)	3.68	0.80	3.04	1.31	5.78	1.78	n.d.	n.d.	n.d.	n.d.	16.40	[54]
Leaves of Green tea (*Camellia sinensis*)	3.06	0.64	1.85	1.20	5.51	0.96	n.d.	n.d.	n.d.	n.d.	13.23	[54]
Leaves of Artichoke (*Cynara scolymus*)	0.08	0.01	5.97	0.12	2.86	0.095	n.d.	n.d.	n.d.	n.d.	9.16	[54]
Leaves and Thalli of Arnica (*Arnicaeflos*)	0.24	0.26	2.80	1.36	2.89	1.40	n.d.	n.d.	n.d.	n.d.	8.98	[54]
Leaves of Rosemary (*Rosmarinus officinalis*)	0.04	0.01	0.005	n.d.	0.12	8.46	n.d.	n.d.	n.d.	n.d.	8.95	[54]
*Coffee* spp.	4.8–5.5	7.1–7.8	52.0–54.2	n.d.	8.1–8.8	4.1–4.8	3.8–4.2	0.005–0.55	0.01–0.26	0.14–1.84	29.5–70.5	[52,63]

n.d.—no data presented.

**Table 3 antioxidants-11-02427-t003:** Marine sources of 5-CQA.

Source	5-CQA Concentration	Ref.
Cyanobacteria		
*Nostoc commune*	2.16 µg/g DW	[65]
*Nostoc* 2S9Bn	9.55 µg/g DW	[66]
Algae		
*Euglena cantabrica*	78 µg/g DW	[65]
*Spongiochloris platensis*	72.11 ng/g	[64]
*Spongiochloris spongiosa*	260 ng/g	[67]
	95.87 ng/g	[64]
*Anabaena doliolum*	82 ± 6.6 ng/g	[67]
*Porphyra tenera*	19 ± 11.9 ng/g	[67]
*Undaria pinnatifia*	10 ± 11.9 ng/g	[67]

**Table 4 antioxidants-11-02427-t004:** Yields HCQAs extracted from plants and wastes.

HCQAs	Plant Type and Part	Extraction Method	Yield	Ref.
3-CQA	Mulberry leaves (*Morus alba* L.)	UAE	0.47 mg/mL	[89]
4-CQA			1.29 mg/mL	
5-CQA			0.65 mg/mL	
5-CQA	Spent coffee grounds	SLE	0.04–0.2 g/L extract	[77]
5-CQA	Silver skin from coffee	SLE	3%	[60]
5-CQA	Honeysuckle (*Lonicera japonicae*)	UAE	37.78 mg/g	[79]
5-CQA	Honeysuckle (*Lonicera japonica*) flower buds	SLE	23.08 mg/kg raw material	[82]
1,4-diCQA			0.32 mg/kg raw material	
3,4-diCQA			42.46 mg/kg raw material	
4,5-diCQA			14.62 mg/kg raw material	
3,4,5-triCQA			4.62 mg/kg raw material	
5-CQA	Sunflower (*Helianthus annuus*) cake	MAE	8.4 mg /g	[78]
5-CQA	Burdock (*Arctium lappa*) leaves	PLE	18.453 (g/kg extract)	[83]
		SFE	8.765(g/kg extract)	
5-CQA	Sunflower (*Helianthus annuus*) seed kernels	SFE	9.06 mg/g raw material	[80]
5-CQA	Honeysuckle (*L. japonica*) flowers	EAE	~4%	[81]
5-CQA	Hardy rubber tree (*Eucommia ulmoides*) leaves	IL-EAE	~3–5.5 mg/g raw material	[75]
5-CQA	Bog bilberry (*Vaccinium uliginosum*) leaves	MSDDE	17.02 mg/g raw material	[90]
5-CQA	Coffee chaff	MMM	0.64–0.94 mg/g raw material	[91]
5-CQA	Coffee pulp	SLE-SSF	600 mg/kg raw material	[92]
3,4,5-triCQA 3,4-diCQA 3,5-diCQA 4,5-diCQA 3-CQA 3-C-FQA	Sweet potato (*Ipomoea batatas*) peels	UAE	n.d.	[93]
3-CQA 4-CQA 5-CQA	Tobacco (*Nicotiana tabacum*) waste	SLE	n.d.	[94]
5-CQA	Tobacco (*Nicotiana tabacum*) waste	UAE	0.497%	[85]
5-CQA	Carrot (*Daucus carota*) pomace	UAE	17.58 μg/g	[86]
5-CQA	Pomegranate (*Punica granatum*) peels	SLE UAE IAE	301–1220 μg/g DW	[71]
			284–1556 μg/g DW	
			679–1562 μg/g DW	
4-CQA	Fennel *(Foeniculum vulgare*) bulbs waste	PLE	1.949 mg/g DW	[87]
3,4-diCQA			0.490 mg/g DW	
1-CQA	Potatoes (Fontane) by-products	UAE	0.36 mg/g DW	[88]
5-CQA			3.04 mg/g DW	
4-CQA			0.39 mg/g DW	
5-FQA			0.05 mg/g DW	
3,4-diCQA			0.09 mg/g DW	
3,5-diCQA			0.40 mg/g DW	
4,5-diCQA			0.16 mg/g DW	

n.d.—no data presented.

**Table 5 antioxidants-11-02427-t005:** The highest yields of other HCQAs obtained by synthetic chemistry methods.

HCQA	Synthesis Method	Total Yield, %	Ref.
5-CQA	Esterification	65	[127]
5-CQA	Esterification	35	[130]
1-CQA 3-CQA 4-CQA	Esterification	41 60 36	[126]
5-FQA	Condensation	19	[129]
3-FQA 4-FQA 5-FQA	Esterification	32.59 14.47 45.10	[131]
1-*p*-CoQA 3-*p*-CoQA 4-*p*-CoQA 5-*p*-CoQA	Esterification	34.47 14.64 14.64 26	[132]
5-SQA	Esterification	15	[133]
3,4-diCQA	Esterification	n.d. ***	[134]
3,4,5-triCQA	Esterification	~14	[135]
1,3,5-triCQA	Esterification	11.71	[136]
3,5-diCQA 3,5-diFQA	Condensation	20.46 21.66	[137]

* n.d.—no data-not enough data presented.

**Table 6 antioxidants-11-02427-t006:** The average prices for the chemical substances and catalysts required for the HCQAs synthesis (Prices are listed from [138].

Compound (Purity 95–99%)	Average Price for 1 L or 1 kg, USD $
Piperidine	80–360
DMAP	60–70
Pyridine	40–150
p-Hydroxybenzaldehyde	50–120
Protocatechuic aldehyde	150–200
Vanillin	100–200
Syringaldehyde	750–1800
*p-*Coumaric acid	940–1500
Vanillic acid	339–512
Syringic acid	586–1100
Caffeic acid	595–633
Quinic acid	8–2920 *

* Depending on the total amount purchased.

**Table 7 antioxidants-11-02427-t007:** Comparison of HCQAs production cases in non-modified micro-organisms.

Microorganism	Substrate	Product	Product Concentration/Titer	Fermentation Time (Days)	Ref.
*Streptomyces albogriseolus* KF977548 (strain AOB)	Coniferyl alcohol or caffeic acid	1-CQA or 3,4,5-triCQA	n.d.	n.d.	[144]
*Penicillium flavigenum* (CML2965)	PD broth	5-CQA	0.38 g/L extract	7	[141]
*Screlotium rolfsii*	Czapek Yeast extract Broth (CYB)		n.d.	5	[142]
*Lodderomyces elongisporus* S216	Modified PD medium		23.39 mg /L	5	[139]
*Sphingomonas yabuuchiae* N21	Modified Beef Extract (BEA) medium		13.04 mg/ L	1	[139]
*Bacillus badius*	Modified Beef Extract (BEA) medium		5.43 mg/ L	1	[139]

n.d.—no data presented.

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
