# Peer review of "Advances in Production of Hydroxycinnamoyl-Quinic Acids: From Natural Sources to Biotechnology"

_antioxidants, 2022, doi:10.3390/antiox11122427_

Round 1
Reviewer 1 Report
The manuscript “Advances in production of hydroxycinnamoyl-quinic acids: from natural sources to biotechnology” was submitted to Antioxidants for publication.
Broad comments:
The topic of the review article is interesting and important, as hydroxycinnamoyl-quinic acids are of importance to nutraceutical industry. In my opinion, the chosen structure and subdivision of the review are appropriate and well investigated.
However, what is a shortcoming of the article, is its many inconsistencies and formal errors that need to be addressed before publication.
These comprise e.g. compound names, especially when using p- for para-. Here p- sometimes is written in italics and sometimes not. Moreover, there is no consistency in using small or big letters for compound names, as shown in Fig. 1 (5-Caffeoylquinic acid vs. 3,5-dicaffeoylquinic acid). In English language, usually small letters are preferred when after a number or letter (e.g. O). However, it is more important to follow a clear naming. Also check for spelling mistakes (e.g. cafeoyl instead of caffeoyl). I am, furthermore, not sure if you should use isochlorogenic acid as a name. It is i) not really used and ii) if used also referred to as isochlorogenic acid A. Thus, staying with 3,5-dicaffeoy-quinic acid makes more sense and also does not lead to confusions as all other chlorogenic acids are monocaffeoyl derivatives.
Please also do not use abbreviations for the genus name of microorganisms. I know that this is quite common and makes sense when only a few microorganisms are discussed. But as you mention quite a number of different organisms it is hard to figure out which one is meant. Therefore, please always write the entire species name (with the only exception being E. coli, which is unambiguous).
In addition, check al tables for consistency. In example, in table 3 different writings are used for the studies drugs. However, it is the task when writing a review article, to decipher these names and make them uniform for the reader. Mixing, Honeysuckle with L. japonica flower and Flos Lonicera japonicae is exactly the way it should not be. The same accounts for Table 6, this time for the names of the depicted microorganisms.
Please also reduce lines 203 to 231 as they repeat what is depicted in table 3 and lines 335 to 368 mentioning the content of table 6.
Please also check for the section numbering. After 2.1, 2.2.1 and 2.2.2 appear, before continuing with 2.2.
As a last major point I would not show glucose as the starting molecule in the figure 3 as that might lead to confusions (it doesn’t start with glucose) and also in parts in figure 6.
Specific comments:
Line 61: Please correct “hydroxycinamoyl” to “hydroxycinnamoyl”.
Line 64: Please correct “except for the some cider yeast” to “except for some cider yeasts”.
Line 69: Please write “resistance against some viruses” instead of “resistance of some viruses”.
Line 359: “Sensu Stricto” should be written in small letters.
Reviewer 2 Report
The manuscript antioxidants-2010626 reviewed the advances in production of hydroxycinnamoyl-quinic acids. The contribution is interesting, well structured, and presented.
Some minor remarks:
Representative pictures of some natural sources of hydroxycinnamoyl-quinic acids (HCQAs) would be attractive.
Figure 2, showing the number of publications dedicated to HCQAs research over the last 10 years, would be better presented by a bar graph. Bar graphs are an effective way to compare items between different groups. They allow the reader to recognize patterns or trends between other groups far more quickly than looking at a line graph.
Figure 4, presenting general agro-industrial waste treatment possibilities, must be improved; it needs to be more complex and give essential information about HCQAs.
The ‘Summary and Outlook’ section must be improved. It must present the most critical findings and provide an outlook on new areas for future research. Please do not include references in this section.
Round 2
Reviewer 1 Report
Dear authors,
The manuscript looks much better now. I only have a few minor comments.
In Table 4, after some compounds a colon is written. Please delete.
In Table 6, sp. and sp are given behind some of the species. Usually, sp. stands instead of the species name (e.g. Sordariomycetes sp.) and is not needed when the species name is know. Please delete.
Please correct Sordariomycete to Sordariomycetes in line 315 and Nicotiana tobacco to Nicotiana tabacum.
Author Response
Please see enclosed response to reviewer
